# Remember to correct the bias when using deep learning for regression!

## Abstract

When training deep learning models for least-squares regression, we cannot expect that the training error residuals of the final model, selected after a fixed training time or based on performance on a hold-out data set, sum to zero. This can introduce a systematic error that accumulates if we are interested in the total aggregated performance over many data points. We suggest adjusting the bias of the machine learning model after training as a default post-processing step, which efficiently solves the problem. The severeness of the error accumulation and the effectiveness of the bias correction are demonstrated in exemplary experiments.

## 1   Problem statement

We consider regression models $f : \mathbb{X} \to \mathbb{R}^d$ of the form

$$f_{\boldsymbol{\theta}}(x) = \boldsymbol{a}^{\mathrm{T}} h_{\boldsymbol{w}}(x) + b \tag{1}$$

with parameters $\boldsymbol{\theta} = (\boldsymbol{w}, \boldsymbol{a}, b)$ and $x \in \mathbb{X}$. Here $\mathbb{X}$ is some arbitrary input space and w.l.o.g. we assume $d = 1$. The function $h_{\boldsymbol{w}} : \mathbb{X} \to \mathbb{R}^F$ is parameterized by $\boldsymbol{w}$ and maps the input to an $F$-dimensional real-valued feature representation, $\boldsymbol{a} \in \mathbb{R}^F$, and $b$ is a scalar. If $\mathbb{X}$ is a Euclidean space and $h$ the identity, this reduces to standard linear regression. However, we are interested in the case where $h_{\boldsymbol{w}}$ is more complex. In particular,

- $f_{\boldsymbol{\theta}}$ can be a deep neural network, where $\boldsymbol{a}$ and $b$ are the parameters of the final output layer and $h_{\boldsymbol{w}}$ represents all other layers (e.g., a convolutional or point cloud architecture);

- $h : \mathcal{X} \mapsto \mathbb{R}$ can be any regression model (e.g., a random forest or deep neural network) and $f_{\boldsymbol{\theta}}$ denotes $h_{\boldsymbol{w}}$ with an additional wrapper, where $a = 1$ and initially $b = 0$.

In the following, we call $b$ the distinct bias parameter of our model (although $\boldsymbol{w}$ may comprise many parameters typically referred to as bias parameters if $h_{\boldsymbol{w}}$ is a neural network). Given some training data $\mathcal{D} = \{(x_1, y_1), \dots, (x_N, y_N)\}$ drawn from a distribution $p_{\text{data}}$ over $\mathbb{X} \times \mathbb{R}$, we assume that the model parameters $\boldsymbol{\theta}$ are determined by minimizing the mean-squared-error (MSE)

$$\mathrm{MSE}_{\mathcal{D}}(f_{\boldsymbol{\theta}}) = \frac{1}{|\mathcal{D}|} \sum_{(x,y) \in \mathcal{D}}^{N} (y - f_{\boldsymbol{\theta}}(x))^2 \ , \tag{2}$$

potentially combined with some form of regularization. Typically, the goal is to achieve a low expected   NEW error $\mathrm{MSE}(f_{\boldsymbol{\theta}}) = \mathbb{E}_{(x,y) \sim p_{\text{data}}}[(y - f_{\boldsymbol{\theta}}(x))^2] = \mathbb{E}[\mathrm{MSE}_{\mathcal{D}_{\text{test}}}(f_{\boldsymbol{\theta}})]$, where the second expectation is over all test data sets drawn i.i.d. based on $p_{\text{data}}$. However, here we are mainly concerned with applications where the (expected) *absolute total error* defined as the absolute value of the sum of residuals

$$\Delta_{\mathcal{D}_{\text{test}}}(f_{\boldsymbol{\theta}}) = \left| \sum_{(x,y) \in \mathcal{D}_{\text{test}}} (y - f_{\boldsymbol{\theta}}(x)) \right| \tag{3}$$

is of high importance. That is, we are interested in the total aggregated performance over many data points. NEW
A related error measure is the *relative total error* given by

$$\delta_{\mathcal{D}_{\text{test}}}(f_{\boldsymbol{\theta}}) = \frac{\Delta_{\mathcal{D}_{\text{test}}}(f_{\boldsymbol{\theta}})}{\left| \sum_{(x,y)\in\mathcal{D}_{\text{test}}} y \right|} \quad , \tag{4}$$

which is similar to the *relative systematic error* $\frac{100}{|\mathcal{D}_{\text{test}}|} \sum_{(x,y)\in\mathcal{D}_{\text{test}}} \frac{y - f_{\boldsymbol{\theta}}(x)}{y}$ (in %, e.g., Jucker et al., 2017)
and the mean error

$$\text{ME}_{\mathcal{D}_{\text{test}}}(f_{\boldsymbol{\theta}}) = \frac{\Delta_{\mathcal{D}_{\text{test}}}(f_{\boldsymbol{\theta}})}{|\mathcal{D}_{\text{test}}|} = \left| \frac{1}{|\mathcal{D}_{\text{test}}|} \sum_{(x,y)\in\mathcal{D}_{\text{test}}} (y - f_{\boldsymbol{\theta}}(x)) \right| . \tag{5}$$

The measures defined by equation 3 to equation 5 are used to quantify the *prediction bias* of the model, that NEW
is, how well $\sum_{(x,y)\in\mathcal{D}_{\text{test}}} f_{\boldsymbol{\theta}}(x)$ approximates $\sum_{(x,y)\in\mathcal{D}_{\text{test}}} y$ for a test set $\mathcal{D}_{\text{test}}$. For $|\mathcal{D}_{\text{test}}| \to \infty$ a constant
model predicting $\hat{y} = \mathbb{E}_{(x,y)\sim p_{\text{data}}}[y]$ would minimize $\Delta_{\mathcal{D}_{\text{test}}}(f_{\boldsymbol{\theta}})/|\mathcal{D}_{\text{test}}|$. However, in practice $\frac{1}{|\mathcal{D}|}\sum_{(x,y)\in\mathcal{D}} y$
and $\frac{1}{|\mathcal{D}_{\text{test}}|}\sum_{(x,y)\in\mathcal{D}_{\text{test}}} y$ can be considerably different from each other and from $\hat{y}$ because of finite sample
effects and violations of the i.i.d. assumption (e.g., due to covariate shift or sample selection bias), so
optimization of individual predictions (e.g., minimizing equation 2) is preferred.

Our study is motivated by applications in large-scale ecosystem monitoring such as convolutional neural
network-based systems estimating tree canopy area from satellite imagery (Brandt et al., 2020) applied for
assessing the total tree canopy cover of a country and learning systems trained on small patches of 3D
point clouds to predict the biomass (and thus stored carbon) of large forests (Jucker et al., 2017; Oehmcke
et al., 2021). For recent reviews of regression methods including deep neural networks for biomass prediction NEW
we refer to Zhang et al. (2020), where also the root of the MSE as well as the mean error are considered
as evaluation criteria and Morais et al. (2021). However, there are many other application areas in which
accumulated predictions matter, such as estimating the overall performance of a portfolio based on estimates
of the performance of the individual assets or overall demand forecasting based on forecasts for individual
consumers.

At a first glance, it seems that low $\mathbb{E}[\text{MSE}_{\mathcal{D}_{\text{test}}}(f_{\boldsymbol{\theta}})]$ guarantees low $\mathbb{E}[\Delta_{\mathcal{D}_{\text{test}}}(f_{\boldsymbol{\theta}})]$, where the expectations are
again with respect to data sets drawn i.i.d. based on $p_{\text{data}}$. Obviously, $\text{MSE}_{\mathcal{D}}(f_{\boldsymbol{\theta}}) = 0$ implies $\Delta_{\mathcal{D}}(f_{\boldsymbol{\theta}}) = 0$ for
any data set $\mathcal{D}$. More general, optimal parameters $\boldsymbol{\theta}^*$ minimizing $\text{MSE}_{\mathcal{D}}(f_{\boldsymbol{\theta}})$ result in $\Delta_{\mathcal{D}}(f_{\boldsymbol{\theta}^*}) = 0$. Actually,
$\frac{\partial \text{MSE}_{\mathcal{D}}(f_{\boldsymbol{\theta}})}{\partial b} = 0$ is necessary and sufficient for the error residuals to sum to zero and thus $\Delta_{\mathcal{D}}(f_{\boldsymbol{\theta}}) = 0$. This NEW
well known fact can directly be seen from equation 7 below. However, if the partial derivative of the error NEW
with respect to $b$ is not zero, a low $\text{MSE}_{\mathcal{D}}(f_{\boldsymbol{\theta}})$ may not imply a low $\Delta_{\mathcal{D}}(f_{\boldsymbol{\theta}})$. In fact, if we are ultimately
interested in the total aggregated performance over many data points, a wrongly adjusted parameter $b$ may
lead to significant systematic errors. Assume that $f^*$ is the Bayes optimal model for a given task and that
$f_\delta$ is the model where the optimal bias parameter $b^*$ is replaced by $b^* - \delta_b$. Then for a test set $\mathcal{D}_{\text{test}}$ of
cardinality $N_{\text{test}}$ we have

$$\sum_{(x,y)\in\mathcal{D}_{\text{test}}} (y - f_{\delta_b}(x)) = N_{\text{test}} \cdot \delta_b + \sum_{(x,y)\in\mathcal{D}_{\text{test}}} (y - f^*(x)) . \tag{6}$$

That is, the errors $\delta_b$ accumulate and, thus, even a very small $\delta_b$ can have a drastic effect on aggregated NEW
quantities. While one typically hopes that errors partly cancel out when applying a model to a lot of data
points, the aggregated error due to a badly chosen bias parameter increases. This can be a severe problem
when using deep learning for regression, because in the canonical training process of a neural network for
regression minimizing the (regularized) MSE the partial derivative of the error w.r.t. the parameter $b$ of the NEW
final model cannot be expected to be zero:

- Large deep learning systems are typically not trained until the partial derivatives of the error w.r.t the NEW
  model parameters are close to zero, because this is not necessary to achieve the desired performance
  in terms of MSE and/or training would take too long.

- The final weight configuration is often picked based on the performance on a validation data set (e.g., Prechelt, 2012), not depending on how close the parameters are to a local optimum as measured, for example, by the maximum norm of the gradient.  NEW

- Mini-batch learning introduces a random effect in the parameter updates, and therefore in the bias parameter value in the finally chosen network.

Thus, despite low MSE, the performance of a (deep) learning system in terms of the total error as defined in equation 3 can get arbitrarily bad. For example, in the tree canopy estimation task described above, you may get a decently accurate biomass estimate for individual trees, but the prediction over a large area (i.e., the quantity you are actually interested in) could be very wrong.

Therefore, we propose to adjust the bias parameter after training a machine learning model for least-squares regression as a default post-processing step. This post-processing can be regarded as playing a similar role as  NEW
model calibration in classification (e.g. Guo et al., 2017). In the next section, we show how to simply compute this correction that exactly removes the prediction bias on the training data (or a subset thereof) and discuss the consequences. Section 3 presents experiments demonstrating the problem and the effectiveness of the proposed solution.

## 2 Solution: Adjusting the bias

NEW

If the sum of residuals on the training data set $\mathcal{D}$ does not vanish, $\Delta_{\mathcal{D}}(f_{\boldsymbol{\theta}}) > 0$, we can also not expect that the residuals will cancel each other on some test set $\mathcal{D}_{\text{test}}$, showing a systematic error leading to a large $\Delta_{\mathcal{D}_{\text{test}}}(f_{\boldsymbol{\theta}})$. Thus, we suggest to apply the minimal change to the model that leads to $\Delta_{\mathcal{D}}(f_{\boldsymbol{\theta}}) = 0$, namely minimizing the MSE on $\mathcal{D} = \{(x_1, y_1), \ldots, (x_N, y_N)\}$ w.r.t. $b$ while fixing all other model parameters $\boldsymbol{w}$ and $\boldsymbol{a}$. For the resulting bias parameter $b^*$ the first derivative w.r.t. $b$ vanishes

$$\left. \frac{\partial \text{MSE}_{\mathcal{D}}(f_{\boldsymbol{\theta}})}{\partial b} \right|_{b=b^*} = \frac{2}{N} \sum_{i=1}^{N} (y_i - \boldsymbol{a}^{\text{T}} h_{\boldsymbol{w}}(x_i) - b^*) = 0 \tag{7}$$

implying $\Delta_{\mathcal{D}}(f_{(\boldsymbol{w}, \boldsymbol{a}, b^*)}) = 0$. Thus, for fixed $\boldsymbol{w}$ and $\boldsymbol{a}$ we can simply solve for the new bias parameter:

$$b^* = \frac{\sum_{i=1}^{N}(y_i - \boldsymbol{a}^{\text{T}} h_{\boldsymbol{w}}(x_i))}{N} = \frac{\sum_{i=1}^{N} y_i - \sum_{i=1}^{N} \boldsymbol{a}^{\text{T}} h_{\boldsymbol{w}}(x_i)}{N} = \underbrace{\frac{\sum_{i=1}^{N} y_i - \sum_{i=1}^{N} f_{\boldsymbol{\theta}}(x_i)}{N}}_{\delta_b} + b \tag{8}$$

In practice, we can either replace $b$ in our trained model by $b^*$ or add $\delta_b$ to all model predictions. The costs of computing $b^*$ and $\delta_b$ are the same as computing the error on the data set used for adjusting the bias. The proposed post-processing step can be related to an algorithm for updating models using additional  NEW
labelled data (e.g., in a transfer learning setting) described by Rodgers et al. (2007), see the discussion in the appendix.

The trivial consequences of this adjustment are that the MSE on the training data set is reduced and  NEW
the residuals on the training set cancel each other. But what happens on unseen data? The model with
$\Delta_{\mathcal{D}}(f_{(\boldsymbol{w}, \boldsymbol{a}, b^*)}) = 0$ can be expected to have a lower $\Delta_{\mathcal{D}_{\text{test}}}(f_{(\boldsymbol{w}, \boldsymbol{a}, b^*)})$ on a test set $\mathcal{D}_{\text{test}}$ than a model with
$\Delta_{\mathcal{D}}(f_{\boldsymbol{\theta}}) > 0$. The effect on the $\text{MSE}_{\mathcal{D}_{\text{test}}}$ is expected to be small. Adjusting the single scalar parameter $b$
based on a lot of data is very unlikely to lead to overfitting. On the contrary, in practice we are typically observing a reduced MSE on external test data after adjusting the bias. However, this effect is typically minor. The weights of the neural network and in particular the bias parameter in the final linear layer are learned sufficiently well so that the MSE is not significantly degraded because the single bias parameter is not adjusted optimally – and that is why one typically does not worry about it although the effect on the absolute total error may be drastic.

**Which data should be used to adjust the bias?** While one could use an additional hold-out set for the final optimization of $b$, this is not necessary. Data already used in the model design process can be used,

because assuming a sufficient amount of data selecting a single parameter is unlikely to lead to overfitting. If there is a validation data set (e.g., for early-stopping), then these data could be used. If data augmentation is used, augmented data sets could be considered. We recommend to simply use all data available for model building (e.g., the union of training and validation set). This minimizes the prediction bias of the model in the same way as standard linear regression. Using a large amount of data for the (typically very small) adjustment of a single model parameter that has no non-linear influence on the model predictions is extremely unlikely to lead to overfitting (as empirically shown in the experiments below), and the more data are used to compute the correction the more accurate it can be expected to be.

**How to deal with regularization?** So far, we just considered empirical risk minimization. However, the bias parameter can adjusted regardless of how the model was obtained. This includes the use of early-stopping (Prechelt, 2012) or regularized risk minimization with an objective of the form $\frac{1}{N}\sum_{i=1}^{N}(y_i - f_{\boldsymbol{\theta}}(x_i))^2 + \Omega(\boldsymbol{\theta})$. Here, $\Omega$ denotes some regularization depending on the parameters. This includes weight-decay, however, typically this type of regularization would not consider the bias parameter $b$ of a regression model anyway (e.g., Bishop, 1995, p. 342).

**Why not adjust more parameters?** The proposed post-processing serves a very well defined purpose. If the error residuals do not sum to zero on the training data set, the residuals on test data can also not be expected to do so, which leads to a systematic prediction error. The proposed adjustment of $b$ is the minimal change to the model that solves this problem. We assume that the model before the post-processing shows good generalization performance in terms of MSE, so we want to change it as little as possible. As argued above and shown in the experiments, just adjusting $b$, which has no non-linear effect on the predictions, based on sufficient data is unlikely to lead to overfitting. On the contrary, in practice an improvement of the generalization performance (e.g., in terms of MSE) is often observed (see also the experiments below).

Of course, there are scenarios where adjusting more parameters can be helpful. For example, it is straightforward to also adjust the factor $a$ in the wrapper such that the partial derivative of the MSE with respect to $a$ vanishes. This has the effect that afterwards the residuals and training inputs are uncorrelated. However, minimizing the unregularized empirical risk w.r.t. many parameters (in particular if we have non-linear effects) bears the risk of overfitting.

## 3 Examples

In this section, we present experiments that illustrate the problem of a large total error despite a low MSE and show that adjusting the bias as proposed above is a viable solution. We start with a simple regression task based on a UCI benchmark data set (Dua & Graff, 2017), which is easy to reproduce (see supplementary material). Then we move closer to real-world applications and consider convolutional neural networks for ecosystem monitoring.

### 3.1 Gas turbine emission prediction

First, we look at an artificial example based on real-world data from the UCI benchmark repository (Dua & Graff, 2017), which is easy to reproduce. We consider the *Gas Turbine CO and NOx Emission Data Set* (Kaya et al., 2019), where each data point corresponds to CO and NOx (NO and $NO_2$) emissions and 9 aggregated sensor measurements from a gas turbine summarized over one hour. The typical tasks are to predict the hourly emissions given the sensor measurements. Here we consider the fictitious task of predicting the total amount of CO emissions for a set of measurements.

**Experimental setup.** There are $36\,733$ data points in total. We assumed that we know the emissions for $N_{\text{train}} = 21\,733$ randomly selected data points, which we used to build our models.

We trained a neural network with two hidden layers with sigmoid activation functions having 16 and 8 neurons, respectively, feeding into a linear output layer. There were shortcut connections from the inputs to the output layer. We randomly split the training data into $16\,733$ examples for gradient computation and 5000 examples for validation. The network was trained for 1000 epochs using Adam (Kinga & Ba, 2015) with

Table 1: Results for the total CO emissions prediction tasks for the different models, where "linear" refers to linear regression, "not corrected" to a neural network without bias correction, and "corrected" to the same neural network with corrected bias parameter. The results are based on 10 trials. The mean and standard error (SE) are given; values are rounded to two decimals; $R^2$, $\Delta$, $\delta$, and ME denote the coefficient of determinations, the absolute total error, the relative error, and the mean error; $\delta$ is given in percent; $\mathcal{D}$ and $\mathcal{D}_{\text{test}}$ refer to data available for model development and testing, respectively.

| MODEL | $R^2_{\mathcal{D}}$ | $R^2_{\mathcal{D}_{\text{test}}}$ | $\Delta_{\mathcal{D}}$ | $\Delta_{\mathcal{D}_{\text{test}}}$ | $\delta_{\mathcal{D}_{\text{test}}}$ | $\text{ME}_{\mathcal{D}_{\text{test}}}$ |
|---|---|---|---|---|---|---|
| linear | 0.56 ±0.0 | 0.57 ±0.0 | 0 ± 0 | 173 ±14 | 0.49 ±0.04 | 0.02 ±0.0 |
| not corrected | 0.78 ±0.0 | 0.72 ±0.0 | 1018 ±70 | 785 ±53 | 2.21 ±0.15 | 0.05 ±0.0 |
| corrected | 0.78 ±0.0 | 0.72 ±0.0 | 0 ± 0 | 122 ± 6 | 0.34 ±0.02 | 0.01 ±0.0 |

a learning rate of $1 \cdot 10^{-2}$ and mini-batches of size 64. The network with the lowest error on the validation data was selected. For adjusting the bias parameter, we computed $\delta_b$ using equation 8 and all $N_{\text{train}}$ data points available for model development. As a baseline, we fitted a linear regression model using all $N_{\text{train}}$ data points.

We used Scikit-learn (Pedregosa et al., 2011) and PyTorch (Paszke et al., 2019) in our experiments. The input data were converted to 32-bit floating point precision. We repeated the experiments 10 times with 10 random data splits, network initializations, and mini-batch shufflings.

**Results.** The results are shown in Table 1 and Figure 1. The neural networks without bias correction achieved a higher $R^2$ (coefficient of determination) than the linear regression on the training and test data, see Table 1. On the test data, the $R^2$ averaged over the ten trials increased from 0.56 to 0.78 when using the neural network. However, the $\Delta_{\mathcal{D}}$ and $\Delta_{\mathcal{D}_{\text{test}}}$ were much lower for linear regression. This shows that a better MSE does not directly translate to a better total error (sum of residuals).

Correcting the bias of the neural network did not change the networks' $R^2$, however, the total errors went down to the same level as for linear regression and even below. Thus, correcting the bias gave the best of both world, a low MSE for individual data points and a low accumulated error.

Figure 1 demonstrates how the total error developed as a function of test data set size. As predicted, with a badly adjusted bias parameter the total error increased with the number of test data points, while for the linear models and the neural network with adjusted bias this negative effect was less pronounced. The linear models performed worse than the neural networks with adjusted bias parameters, which can be explained by the worse accuracy of the individual predictions.

## 3.2 Forest Coverage

Deep learning holds great promise for large-scale ecosystem monitoring (Persello et al., 2022; Yuan et al., 2020), for example for estimating tree canopy cover and forest biomass from remote sensing data (Brandt et al., 2020; Oehmcke et al., 2021). Here we consider a simplified task where the goal is to predict the amount of pixels in an image that belong to forests given a satellite image. We generated the input data from Sentinel 2 measurements (RGB values) and the accumulated pixels from a landcover map[1] as targets, see Figure 2 for examples. Both, input and target are at the same 10 m spatial resolution, collected/estimated in 2017, and cover the country of Denmark. Each sample is a $100 \times 100$ large image with no overlap between images.

**Experimental setup.** From the 127 643 data points in total, 70% (89 350) were used for training, 10% (12 764) for validation and 20% (25 529) for testing. For each of the 10 trials a different random split of the data was considered.

---

[1]https://www.esa.int/Applications/Observing_the_Earth/Copernicus/Sentinel-2/Land-cover_maps_of_Europe_from_the_Cloud

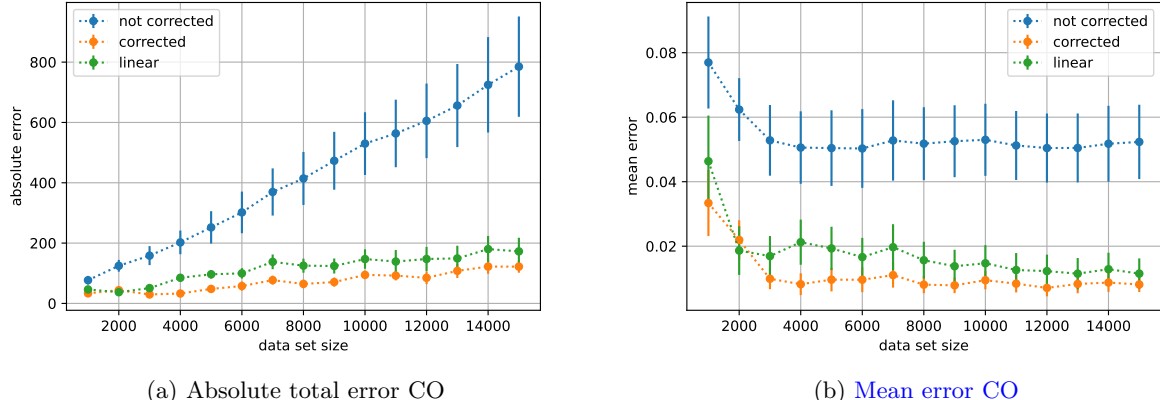

(a) Absolute total error CO            (b) Mean error CO

Figure 1: Absolute errors (absolute value of the summed residuals) are shown on the left and the mean errors on the right for the CO emission prediction task. Both are are presented in relation to the test set size. The error bars show the standard error (SE). Here "linear" refers to linear regression, "not corrected" to a neural network without bias correction, and "corrected" to the same neural network with corrected bias parameter. The results are averaged over 10 trials, the error bars show the standard error (SE).

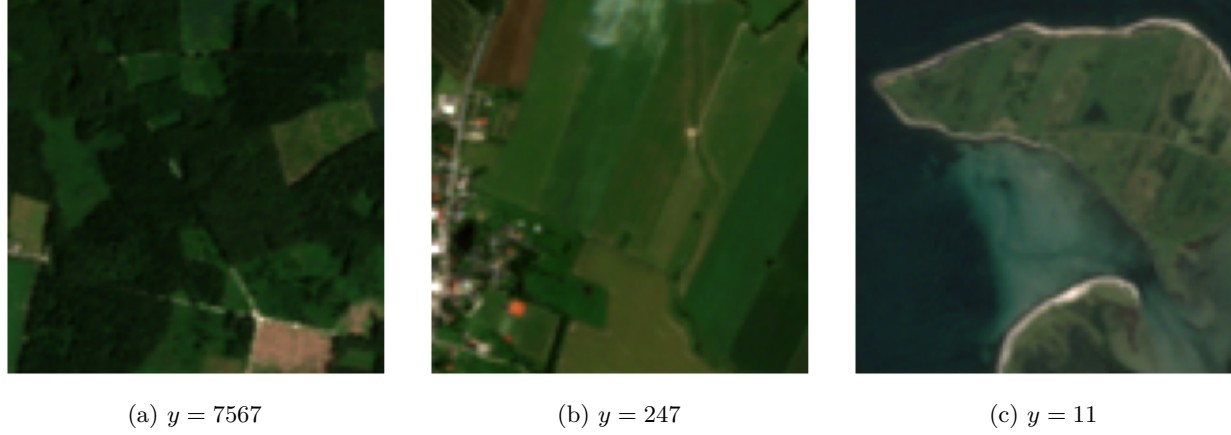

(a) $y = 7567$          (b) $y = 247$          (c) $y = 11$

Figure 2: Exemplary inputs and targets ($y$) for the forest coverage dataset. (a) shows a scene with 75.7%, (b) with 2.5%, and (c) with 0.1% forest.

Table 2: Results of forest coverage prediction, $R^2$ and $\Delta$, $\delta$, and ME denote the coefficient of determinations, absolute total error, relative error, and mean error; $\mathcal{D}$ and $\mathcal{D}_{\text{test}}$ are all data available for model development and testing, respectively. The relative total error $\delta$ is given in percent. Average and standard error (SE) of these metrics are given over 10 trials for the different models, where "mean" refers to predicting the constant mean of the training set, "not corrected" to EfficientNet-B0 without bias correction, and "corrected" to the same neural network with corrected bias parameter. Values are rounded to three decimals.

| **MODEL** | $R^2_{\mathcal{D}}$ | $R^2_{\mathcal{D}_{\text{test}}}$ | $\Delta_{\mathcal{D}}$ | $\Delta_{\mathcal{D}_{\text{test}}}$ | $\delta_{\mathcal{D}_{\text{test}}}$ | $\text{ME}_{\mathcal{D}_{\text{test}}}$ |
|---|---|---|---|---|---|---|
| mean | $0.000 \pm 0.000$ | $-3 \cdot 10^{-5} \pm 0.0$ | $6 \pm 2$ | $169\,666 \pm 48\,944$ | $0.955 \pm 0.272$ | $665 \pm 192$ |
| not corrected | $0.992 \pm 0.027$ | $0.977 \pm 0.0$ | $389\,747 \pm 77\,987$ | $152\,666 \pm 22\,164$ | $0.864 \pm 0.124$ | $598 \pm 87$ |
| corrected | $0.992 \pm 0.027$ | $0.977 \pm 0.0$ | $3 \pm 1$ | $59\,819 \pm 10\,501$ | $0.338 \pm 0.059$ | $234 \pm 41$ |

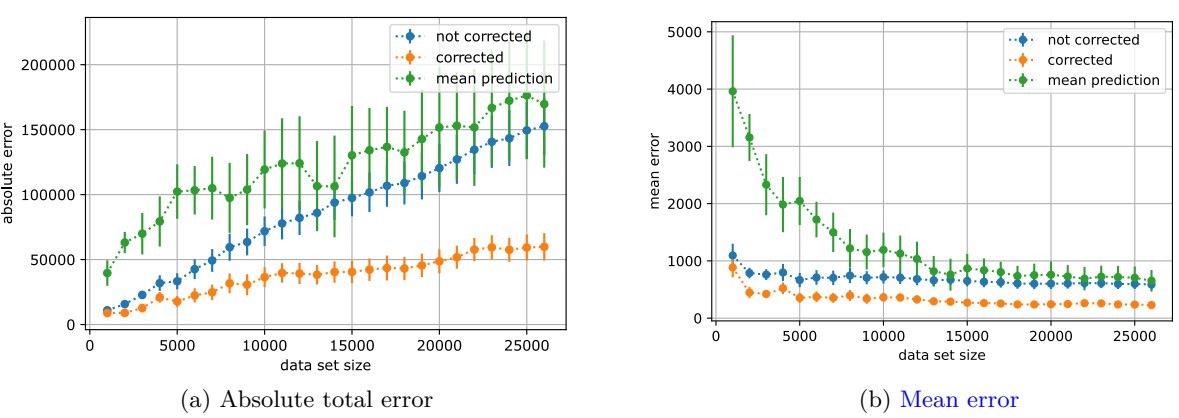

(a) Absolute total error

(b) Mean error

Figure 3: The absolute errors (absolute value of the summed residuals) are shown on the left and the relative errors on the right for the forest coverage prediction task. Both are are presented in relation to the test set size. The error bars show the standard error (SE). Results were averaged over 10 trials and show the different models, where "mean" refers to predicting the constant mean of the training set, "not corrected" to EfficientNet-B0 without bias correction, and "corrected" to the same neural network with corrected bias parameter.

We employed the EfficientNet-B0 (Tan & Le, 2019), a deep convolutional network that uses mobile inverted bottleneck MBConv (Tan et al., 2019) and squeeze-and-excitation (Hu et al., 2018) blocks. It was trained for 300 epochs with Adam and a batch size of 256. For 100 epochs the learning rate was set to $3 \cdot 10^{-4}$ and thereafter reduced to $1 \cdot 10^{-5}$. The validation set was used to select the best model w.r.t. $R^2$. When correcting the bias, the training and validation set were combined. We considered the constant model predicting the mean of the training targets as a baseline.

**Results.** The results are summarized in Figure 3 and Table 2. The bias correction did not yield a better $R^2$ result, with 0.992 on the training set and 0.977 on the test set. However, $\Delta_{\mathcal{D}_{\text{test}}}$ on the test set decreased by a factor of 2.6 from $152\,666$ to $59\,819$. The $R^2$ for the mean prediction is by definition 0 on the training set and was close to 0 on the test set, yet $\Delta_{\mathcal{D}_{\text{test}}}$ is $169\,666$, meaning that a shift in the distribution center occurred, rendering the mean prediction unreliable.

In Figure 3, we show $\Delta_{\mathcal{D}_{\text{test}}}$ and $\delta_{\mathcal{D}_{\text{test}}}$ while increasing the test set size. As expected, the total absolute error of the uncorrected neural networks increases with increasing number of test data points. Simply predicting the mean gave similar results in terms of the accumulated errors compared to the uncorrected model, which shows how misleading the $R^2$ can be as an indicator how well regression models perform in terms of the accumulated total error. When the bias was corrected, this effect drastically decreased and the performance was better compared to mean prediction.

## 4    Conclusions

Adjusting the bias such that the residuals sum to zero should be the default post-processing step when doing least-squares regression using deep learning. It comes at the cost of at most a single forward propagation of the training and/or validation data set, but removes a systematic error that accumulates if individual predictions are summed.

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

## A  Local bias correction

Our bias correction can be related to other post-processing methods. There is a line of research that studies how the output of a model $h$ can be adjusted using an additional labelled data set $\mathcal{D}'$ not used for building the model, and the approach by Rodgers et al. (2007) resembles the recommended bias correction.

The idea is to correct the prediction $h(\boldsymbol{x})$ of the model by the (weighted) mean error of $h$ when applied to the $K$ nearest neighbors of $\boldsymbol{x}$ in $\mathcal{D}'$. Let $(\boldsymbol{x}'_{\boldsymbol{x}:k}, y'_{\boldsymbol{x}:k})$ denote the $K$-th nearest neighbor of $\boldsymbol{x}$ in a data set $\mathcal{D}'$. The distance is measured using a metric $d$, where ties can be broken at randomly or deterministically. Following Bruneau & McElroy (2006) and Rodgers et al. (2007), we consider the Mahalanobis distance $d(\boldsymbol{x}, \boldsymbol{z}) = \sqrt{(\boldsymbol{x} - \boldsymbol{z})^{\mathrm{T}} \boldsymbol{C}^{-1} (\boldsymbol{x} - \boldsymbol{z})}$ for real vectors $\boldsymbol{x}$ and $\boldsymbol{z}$, where $\boldsymbol{C}$ is empirical covariance matrix of the features based on the sample in $\mathcal{D}'$. The output $h(\boldsymbol{x})$ is then corrected to $f(\boldsymbol{x})$ using

$$f(\boldsymbol{x}) = h(\boldsymbol{x}) + \frac{\sum_{k=1}^{K} \omega_{\boldsymbol{x}:k}(h(\boldsymbol{x}'_{\boldsymbol{x}:k}) - y'_{\boldsymbol{x}:k})}{\sum_{i=1}^{K} \omega_{\boldsymbol{x}:k}} \ . \tag{9}$$

Here $\omega_{\boldsymbol{x}:k}$ is a weight depending on $d(\boldsymbol{x}, \boldsymbol{x}'_{\boldsymbol{x}:k})$. The number $K$ of neighbors is a hyperparameter. The term $\frac{\sum_{k=1}^{K} \omega_{\boldsymbol{x}:k} h(\boldsymbol{x}'_{\boldsymbol{x}:k})}{\sum_{i=1}^{K} \omega_{\boldsymbol{x}:k}}$ is the weighted $K$-nearest neighbor prediction for $\boldsymbol{x}$ using $\mathcal{D}'$, and $\frac{\sum_{k=1}^{K} \omega_{\boldsymbol{x}:k} y'_{\boldsymbol{x}:k}}{\sum_{i=1}^{K} \omega_{\boldsymbol{x}:k}}$ can be viewed as the corresponding weighted target. For constant $\omega_{\boldsymbol{x}:k} = 1$, we get $f(\boldsymbol{x}) = h(\boldsymbol{x}) + \frac{1}{K}\sum_{k=1}^{K} h(\boldsymbol{x}'_{\boldsymbol{x}:k}) - \frac{1}{K}\sum_{k=1}^{K} y'_{\boldsymbol{x}:k}$. If we further set $\mathcal{D}' = \mathcal{D}$ and $K = |\mathcal{D}|$ this correction is identical to the suggested bias correction. For smaller $K$, we can think of this method as a *local bias correction*, which adjusts the bias individually for each input based on neighboring training data points.

Our proposed post-processing step efficiently solves the well-defined problem that the error residuals on the training data may not sum to zero. The method suggested – for a different purpose – by Rodgers et al. (2007) is a heuristic with several crucial hyperparameters, obviously $K$ but also the choice of the weighting function for computing the $\omega_{\boldsymbol{x}:k}$. Instead of a one-time correction of a single model parameter, which can be done in linear time, the approach by Rodgers et al. (2007) requires evaluation of a $K$-nearest search in each application of a model, which drastically increases storage and time complexity for training data sets. The performance of their approach depends on the quality of the nearest neighbor regression. Nearest neighbor regression with Mahalanobis distance or standard Euclidean distance is unsuited for image analysis tasks as the one in Section 3.2. The input dimensionality is too high for the amount of training data and neither Mahalanobis distance nor standard Euclidean distance between raw pixels are appropriate to measure image similarity. In contrast, on the artificial problem in Section 3.1 with 9 inputs each representing a meaningful feature, nearest neighbor regression can be expected to work.

We applied the local bias correction to the problem in Section 3.1 with $K = 3$ as suggested by Rodgers et al. (2007). This resulted in $R^2_{\mathcal{D}_{\text{test}}} = 0.73 \pm 0.0$, $\Delta_{\mathcal{D}} 0.0 \pm 0.0$, $\Delta_{\mathcal{D}_{\text{test}}} 51 \pm 5.0$, $\delta_{\mathcal{D}_{\text{test}}} = 0.38\% \pm 0.02\%$, and $\text{ME}_{\mathcal{D}_{\text{test}}} = 0.02 \pm 0.0$. In this toy example, the nearest-neighbor auxiliary model performs very well. Still, while the bias correction reduced the systematic errors compared to the uncorrected neural network, it performed worse than the proposed rigorous post-processing (see Table 1).

