# OpenReview forum: "Remember to correct the bias when using deep learning for regression!"
_TMLR — Rejected by TMLR_

### Review · Reviewer_ojpd · 2022-04-14

**Summary Of Contributions:**

The basic formal setting of this paper is regression using the quadratic loss using a model that is a linear combination of potentially non-linear features (with weights $\mathbf{a}$ and intercept $b$), where the nature of the non-linear features is also to be determined as part of the learning process as a whole (determined by $\mathbf{w}$).

While the formal setting is regression under the quadratic loss, the authors' main interest as I understand it is on the "bias" that tends to occur when solving jointly in $\mathbf{\theta}$ and stopping before reaching a stationary point, where bias refers to (signed) residuals with a non-zero empirical mean. In practice, this means the learned candidate may tend to over-shoot or under-shoot the target variable.

The main methodological contribution here is the suggestion to shift the learned candidate by the average residual to correct for this bias on the training data. The authors provide evidence for the utility of this approach in the form of numerical experiments using real data in which they find that their correction does not hurt the test performance under the quadratic loss, while maintaining smaller residual sums than are obtained without the correction.

**Broader Impact Concerns:**

No issues of broader impact.

**Requested Changes:**

I ask that the authors consider my feedback under strengths and weaknesses, and reconsider exactly what the research problem is that they are trying to solve. Once a real, concrete problem is made clear, a broader look at potential solution methods, and a search for convincing evidence that the problem has been solved to some degree will be the next step. I personally think it is somewhat difficult to accept this paper in its present form, since without a convincing problem, it is difficult to evaluate potential solutions.

**Strengths And Weaknesses:**

*Strengths:*

The paper is well-structured, written in a concise but inviting tone, and is easy to follow at a high level, with the basic problem setup and proposed procedure described clearly. The experimental setup and results are also communicated effectively.

*Weaknesses:*

Overall, I feel the paper is quite vague in terms of what exactly the *problem* faced is, from both the perspectives of numerical optimization and statistical learning. Let me raise a few concrete points that I tripped up on when reading this paper that I think relate to this weakness.

- As long as we are using the quadratic loss and we arrive at a stationary point (i.e., any point $\mathbf{\theta}$ where the gradient is $\nabla \mathcal{R}_{\mathcal{D}}(\mathbf{\theta}) = \mathbf{0}$), then we are guaranteed to have residuals with empirical mean zero, i.e, their (3) and (4) are both zero. The authors seem to be aware of this based on their statements on p.2 ("More general, the optimal parameters minimizing [...] result in [zero sum of residuals]"), so their motivations must be that many real-world settings do not achieve nearly-zero gradients? Again on p.2, they seem to state their basic concerns in three bullets after "...the parameter $b$ of the final model cannot be expected to be optimal," but again I find this rather troublesome. We do not need anything to be *optimal*, we just need the gradient of the MSE objective to be close to zero; since we do not have convexity in general, we can have (globally) sub-optimal parameters (potentially huge MSE) which still have zero residual sums. Is this okay in the mindset of the authors? The authors have not touched on this possibility, it seems.
- Continuing on this same point, one often trains non-linear models (including neural networks) by running an optimization algorithm iteratively until we see a certain degree of numerical convergence. Gradient-based methods are very common, and they typically have the property that numerical convergence basically means the gradient of the MSE is close to zero (although MSE itself may still be large). As such, if the three bullet points near the end of p.2 are meant to suggest that the partial derivative taken with respect to $b$ tends to be large in practice, I find that claim somewhat hard to digest without harder evidence.
- On the point of evidence, the tables in section 3 are comparing MSE (sum divided by number of samples) with plain sums; dividing $\Delta_{\mathcal{D}}$ by the number of samples would likely make these results a lot less remarkable. I think most readers will not be convinced by these results that severe "bias" is a problem inherent in deep learning.

The above points highlight why I found the *research problem* very unclear. Now on the other side, related to the proposed method, while I understand perfectly well why the proposed method of the authors was selected, the perspective here seems way too narrow. Why just ensure $b$ satisfies the first-order necessary conditions for optimality? Why not consider $\mathbf{a}$ as well? We could do a two-stage learning process, fixing $\mathbf{a}$ and $b$ to start with, running a learning algorithm to determine some $\widehat{\mathbf{w}}$, and then solving the ordinary least squares problem in $(\mathbf{a},b)$ conditioned on $\widehat{\mathbf{w}}$. This possibility is not discussed, and I think many readers might expect that it would be beneficial to set aside some data to be used in each of these steps, but the authors dismiss this notion as being "not necessary" (p.3). I think we can easily come up with both settings where this is and is not necessary, so it seems strange to brush this under the carpet without further evidence.

---

### Review · Reviewer_C42P · 2022-04-18

**Summary Of Contributions:**

This paper propose to correct the "bias" term in a regression (e.g. neural networks) model as a post-processing step. This post-processing step fixes all other parameters, and update the bias term to the close-form minimizer of the MSE loss on the training set. This paper argues that this has the advantage that the absolute total error, defined as the absolute value of the sum of residuals, can be reduced on the training set, and this step also tends to have no or small positive impact on model's generalization. Therefore, this should almost always be performed as it is also computationally cheap (equivalent to computing the average loss on the training set).

**Requested Changes:**

Refinements to writing: see "weakness" above.

Minor: The MSE in Eq(2) is using the symbol R, which is very easily confused with the R^2 symbol in the results table indicating the coefficient of determination. Maybe consider using a different symbol for the MSE loss?

**Strengths And Weaknesses:**

Strength: this paper proposed a simple post processing step that effectively solves the bias issue. This method is easy to understand and to implement.

Weakness: even though the paper mentioned some examples where the absolute total error is important. I'm a little bit confused if this is referring to the training set error or the test set error. The proposed method is targeting the error on the training set, but it seems empirically the error on the test set also reduces. It would be great if the paper could provide a concrete motivating example with detailed analyze of the consequences when the (training) absolute total error is not minimized. And also make it a bit more clear when are we talking about training errors and when test errors.

---

### Review · Reviewer_Yjaz · 2022-04-25

**Summary Of Contributions:**

In the paper, the author points out bias issues with deep learning models for regression that may not affect squared mean error but accumulates on summed estimations such as tree canopy estimation. The authors explain how to correct bias with a simple mean error correction on the output of the model's predictions. They then present results corroborating the usefulness of a bias correction on a Gas turbine emission prediction task, modified to estimate total emission instead to demonstrate the bias issue with summed estimations, and Forest coverage task.

**Broader Impact Concerns:**

I see no broader impact concerns.

**Requested Changes:**

I would request one major change which would require rewriting most of the paper and a second round of reviews in my opinion. The bias correction method should be compared to other existing methods explaining how it differs, under which contexts it is preferable, and under which contexts it should be avoided. Looking quickly for bias correction methods I have found [1] (eq. 5) which is equivalent but more general. It uses the mean error weighted by the distance between a given test point and the training points. It seems clear to me that the proposed method does not provide any new insight compared to the one in [1].

The main interest I could see for this paper is bringing attention of the ML community to the issue of bias correction, but this would be better done with a thorough review of bias correction for machine learning regression models.

[1] Rodgers, Sarah L., Andrew M. Davis, Nick P. Tomkinson, and Han van de Waterbeemd. "QSAR modeling using automatically updating correction libraries: application to a human plasma protein binding model." Journal of chemical information and modeling 47, no. 6 (2007): 2401-2407.

**Strengths And Weaknesses:**

Strengths
+ The paper is well written and clear.
+ The method is simple.

Weaknesses
- There is no related work presented. I could hardly believe that this simple correction method has not been mentioned before and easily found similar solutions, see for instance [1]

[1] Rodgers, Sarah L., Andrew M. Davis, Nick P. Tomkinson, and Han van de Waterbeemd. "QSAR modeling using automatically updating correction libraries: application to a human plasma protein binding model." Journal of chemical information and modeling 47, no. 6 (2007): 2401-2407.

---

### Decision · Action_Editors · 2022-06-17

**Recommendation:** Reject

**Comment:**

The claims made in the submission are supported and experiments are confirming these. The post-processing method proposed in this submission is relatively straightforward, but of general interest to the community. The paper also reads well aside from a few remaining typos.

However, the paper is missing a proper related work section. The authors added a paragraph in the main body and discussed the relation to one reference mentioned by one of the reviewers in the Appendix (+ reported new experimental results, which is great), but stopped there. I would encourage the authors to be more thorough in positioning their work with respect to prior work and discussing related work of bias correction in ML in more general terms.